# Cryo-EM of prion strains from the same genotype of host identifies conformational determinants

**Forrest Hoyt**[1◉], **Parvez Alam**[2◉], **Efrosini Artikis**[2◉], **Cindi L. Schwartz**[1], **Andrew G. Hughson**[2], **Brent Race**[2], **Chase Baune**[2], **Gregory J. Raymond**[2], **Gerald S. Baron**[2], **Allison Kraus**[3,4], **Byron Caughey**[2]*

**1** Research Technologies Branch, Rocky Mountain Laboratories, National Institute of Allergy and Infectious Diseases, National Institutes of Health, Hamilton, Montana, United States of America, **2** Laboratory of Persistent Viral Diseases, Rocky Mountain Laboratories, National Institute of Allergy and Infectious Diseases, National Institutes of Health, Hamilton, Montana, United States of America, **3** Department of Pathology, Case Western Reserve University School of Medicine, Cleveland, Ohio, United States of America, **4** Cleveland Center for Membrane and Structural Biology, Case Western Reserve University, Cleveland, Ohio, United States of America

◉ These authors contributed equally to this work.
* bcaughey@nih.gov

**Data Availability Statement:** Cryo-EM density maps and the atomic model of the a22L PrPSc fibrils have been deposited at the Electron Microscopy Data Bank and Protein Data Bank with

## Abstract

Prion strains in a given type of mammalian host are distinguished by differences in clinical presentation, neuropathological lesions, survival time, and characteristics of the infecting prion protein (PrP) assemblies. Near-atomic structures of prions from two host species with different PrP sequences have been determined but comparisons of distinct prion strains of the same amino acid sequence are needed to identify purely conformational determinants of prion strain characteristics. Here we report a 3.2 Å resolution cryogenic electron microscopy-based structure of the 22L prion strain purified from the brains of mice engineered to express only PrP lacking glycophosphatidylinositol anchors [anchorless (a) 22L]. Comparison of this near-atomic structure to our recently determined structure of the aRML strain propagated in the same inbred mouse reveals that these two mouse prion strains have distinct conformational templates for growth via incorporation of PrP molecules of the same sequence. Both a22L and aRML are assembled as stacks of PrP molecules forming parallel in-register intermolecular β-sheets and intervening loops, with single monomers spanning the ordered fibril core. Each monomer shares an N-terminal steric zipper, three major arches, and an overall V-shape, but the details of these and other conformational features differ markedly. Thus, variations in shared conformational motifs within a parallel in-register β-stack fibril architecture provide a structural basis for prion strain differentiation within a single host genotype.

accession codes EMD-28089 and PDB ID 8EFU, respectively.

**Funding:** This work was supported by Intramural Research Program, NIAID, NIH [Project ZIAAI000580-22 (BC) and core funding for the Research Technologies Branch]. All authors, except A.K., received salary from the NIAID. A.K. was supported by Case Western Reserve University, the Britton Fund, and the Clifford V. Harding and Mina K. Chung Professorship in Pathology. The funders had no role in study design, data collection and analysis, decision to publish, or preparation of the manuscript.

**Competing interests:** The authors have declared that no competing interests exist.

## Author summary

Prions are protein-based pathogens that can spread within and between hosts without carrying a pathogen-specific nucleic acid genome. Given this protein-based propagation mechanism, a long-standing mystery in the prion disease field has been the molecular basis of distinct, faithfully propagating strains in a single type of mammalian host. Here we provide a direct, high-resolution cryo-EM-based comparison of the structures of two highly infectious prion strains isolated from the brains of mice of a single genotype. We show in detail how these two prion strains are protein filaments of mouse PrP molecules that display distinct conformational templates for growth on their tips. Our results identify purely conformational, rather than sequence-based, underpinnings of infectious and deadly prion strains.

## Introduction

Prion diseases or transmissible spongiform encephalopathies (TSEs) are fatal infectious neuro-degenerative diseases including CJD (Creutzfeldt-Jakob disease) in humans, bovine spongiform encephalopathy in cattle, scrapie in sheep, and CWD (chronic wasting disease) in cervids [1,2]. Neuropathological hallmarks of prion diseases include spongiform change, neuronal loss, astrocytosis, and accumulation of pathologic forms of the hosts' prion protein (PrP) that, when infectious, have generically been termed PrP$^{Sc}$ for PrP-scrapie [3]. The normal cellular form of PrP (PrP$^C$) typically exists as a glycosylated, glycophosphatidylinositol (GPI)- linked, membrane bound monomer with a largely α-helical C- terminal domain and disordered N-terminal domain [4]. In contrast, PrP$^{Sc}$ is multimeric and when purified, at least, usually takes the form of amyloid fibrils with ordered cores that are high in intermolecular β-sheets and loop structures [5–7]. This aggregated form of PrP is able to grow by refolding PrP$^C$ and incorporating it into an ordered, transmissible assembly.

Given the protein-based mechanism of prion propagation, and the fact that prions do not carry their own mutable nucleic acid genome from host-to-host, one of the most intriguing and long-standing questions in prion biology has been the molecular basis of prion strains. Prion strains are infectious isolates that when transmitted to a given type of host exhibit characteristic clinical phenotypes, neuropathological lesions, survival times, and PrP deposition patterns [8]. Many low-resolution biochemical and spectroscopic comparisons of preparations of different PrP$^{Sc}$ strains have provided evidence of distinct conformations even when strains are formed from the same polypeptide sequence (e.g. [2,9–17]). Moreover, PrP$^{Sc}$ strains have been shown to impose their general conformational attributes onto newly recruited PrP molecules in cell-free conversion and amplification reactions [9,18]. Accordingly, consistent prion strain propagation has been postulated to involve conformational templating by PrP$^{Sc}$ [9,11].

In accordance with this concept are comparisons of the first high-resolution cryo-electron microscopy (cryo-EM) structures of fully infectious brain-derived PrP$^{Sc}$ prions, namely the hamster 263K [6], and mouse wildtype (wt) RML [7] and GPI-anchorless (a) RML [5] prions. Each is an amyloid fibril with a parallel in-register intermolecular β-sheet (PIRIBS)-based core containing amino-proximal (N), middle, and disulfide arches, as well as a steric zipper that holds the extreme N-terminal residues of the core against the head of the middle arch. (Note: we have taken to using the term arch, instead of β-arch, because some cases do not fully meet the β-arch (or arc) criterion of having β-strands on both sides of the arch that interact via their sidechains). However, while wtRML and aRML prion structures are quite similar to one another, they differ markedly from the 263K structure in their conformational details [5–7].

Importantly, the RML and 263K structures also differ in amino acid sequences at 8 residues within their ordered amyloid cores, raising questions about the contributions of sequence differences to their respective templating activities and species tropisms.

To identify purely conformational determinants of prion strains, high-resolution comparisons of strains propagated in hosts of the same genotype are needed. Here we address this central question in prion biology by providing a high-resolution structure of the a22L prion strain isolated from a GPI-anchorless PrP-expressing transgenic mouse strain and compare it to our previously determined structure of the aRML strain propagated in the same genotype of mouse [5]. Previous studies have already suggested conformational differences between these two prion strains. For example, Sim *et al* reported the differences in ultrastructure of a22L and aRML using TEM and AFM [19]. Comparison of HDX- M/S data for a22L and aRML by Smirnovas *et al* showed localized differences in protection against deuterium exchange within their amyloid cores. a22L and aRML also differ in their infrared spectra, most notably in the β-sheet region of the amide I region [15]. Also suggestive of distinct conformations was a report from Bett *et al* that polythiophene acetic acid (PTAA) emission spectra from a22L were more redshifted than those of aRML [20]. They also found that a22L is more resistant to proteinase K (PK) digestion than aRML. In terms of neuropathology, wildtype 22L PrP^Sc deposition occurs mainly in astroglia in several brain regions during early stages of infection while RML PrP^Sc associates with astroglia in the thalamus, cortex, as well as in neurons and neuropil of the substantia nigra and hypothalamus [21]. Detailed knowledge of the conformations of these prion strains should provide a foundation for understanding their respective pathophysiological mechanisms. Our current study specifies, with near atomic resolution, how the a22L strain conformation differs from that of the aRML and wtRML strains of the same murine PrP sequence. These findings reveal purely conformational bases for prion strain differentiation.

## Results

### a22L PrP^Sc purification and infectivity

Protein-stained SDS-PAGE gels and immunoblotting of the a22L preparation that we used for cryo-EM was found to have high purity with respect to PrP content that was indistinguishable from what we have reported previously for other a22L preparations [16]. Intracerebral inoculation of 0.1 μg of the preparation into tga20 mice [22] (n = 4) led to terminal prion disease requiring euthanasia in all recipients with a mean incubation period of 88 ± 8 days. This was indistinguishable from the mean of 88 ± 7 days obtained from inoculation of 1% brain homogenate that, based on prior end-point titration in tga20 mice, contained $3 \times 10^6$ 50% lethal i.c. doses (LD50s) of wildtype 22L scrapie. These results show that the a22L preparation was highly infectious. Given that incubation period correlates inversely with prion titer for a given prion strain in a given type of host [23], the similarity of these incubation periods suggests that the a22L preparation contains roughly $3 \times 10^{10}$ LD50/mg, which is comparable to that of our previously titered aRML preparation [5]. However, a caveat to this a22L titer estimate is that the purified a22L preparation and wildtype 22L in brain homogenate may not dilute out to end point equivalently due, for example, to differences in fibril aggregation state. Nonetheless, inoculations of 10 pg of the a22L preparation caused terminal scrapie in 4/4 mice (126 +/- 4 dpi). Inoculations of 100 fg has led to euthanasia of 2/4 mice at 139 +/0 dpi, but the other recipients of this, and further, dilutions remain alive as of this writing at 150 dpi.

### Negative stain EM and cryo-EM tomographic analyses

Negative stain EM indicated a predominantly fibrillar morphology with a mixture of individual, laterally associated, and crossed fibrils (Fig 1A). To determine handedness of fibril twist

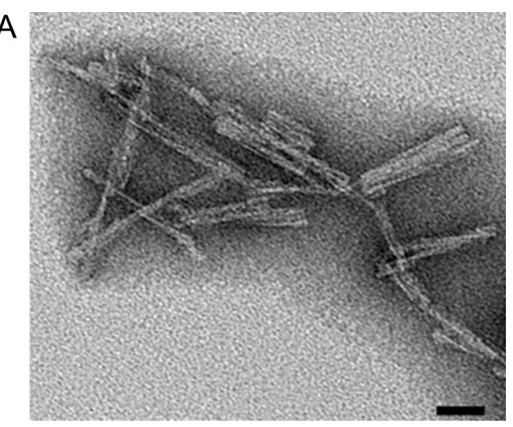

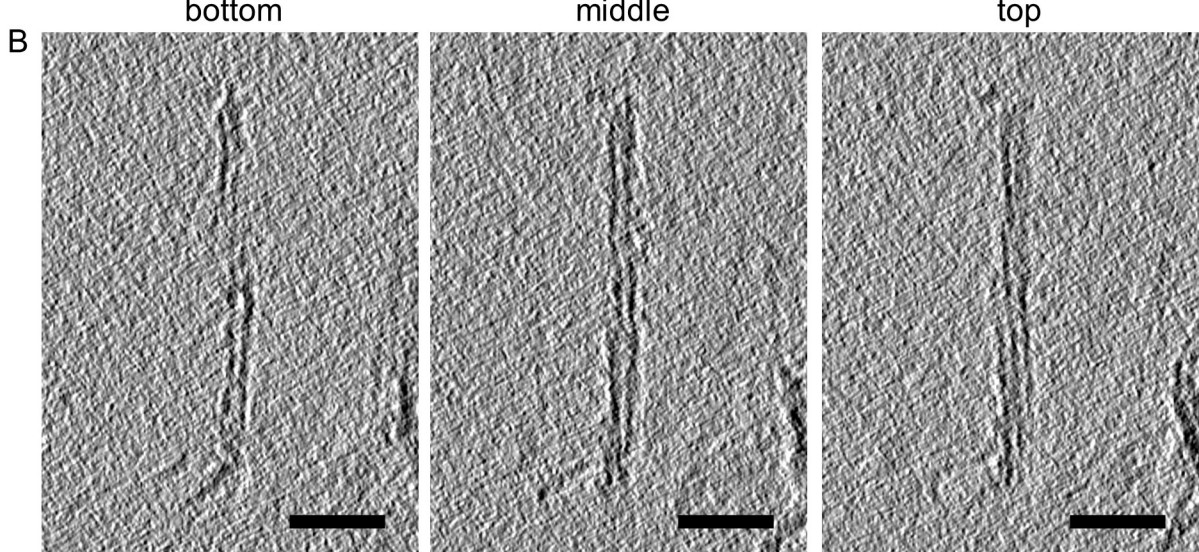

**Fig 1. Negatively stained transmission (A) and cryo-electron tomography (B) of purified a22L prion preparation.** In B, left-handed twist is evident in going from bottom to top tomographic slices through the fibril. Bars = 50 nm.

we performed cryo-electron tomography. As is typical of prion preparations, both individual and bundled fibrils were observed, but their overall relative proportions were impossible to ascertain due to the additional presence of dense, unresolvable mats of fibrils. All a22L fibrils sufficiently isolated for tomographic analysis (n = 41) had a left-handed helical twist (Fig 1B). As was the case for 263K and aRML prion fibril preparations, we also observed globules of unidentified composition along the sides of some of the fibrils.

## Single-particle analysis and 3D image reconstruction

We obtained structural details of a22L prion fibrils using single particle acquisition and helical reconstruction [24] with parameters given in Methods and Table 1. Both individual fibrils of widely varying lengths, and lateral bundles thereof, were observed and those that were not lying on top of one another, regardless of length, were analyzed in the initial 2D images (movies) (Figs 1 and 2A). Less distinct clumps of unknown nature were also sometimes seen, e.g. in lower right hand corner of Fig 2A. Fast Fourier transforms of the image in Fig 2A, for example, indicated regular spacings of ~5.0 Å (Fig 2B, prior to pixel size correction used in 3D

**Table 1. Cryo-EM data, refinement, and validations.**

| Data collection and processing | |
|---|---|
| Magnification | 81,000x |
| Voltage (kV) | 300 |
| Electron dose (e-/ Å$^2$) | 57 |
| Pixel size (Å/pix) | 1.1 |
| Corrected pixel size (Å/pix) | 1.045 |
| Symmetry imposed | C1 |
| Initial particle segments | 226306 |
| Final particle segments | 16409 |
| Map resolution (Å) | 3.2 |
| Helical rise (Å) | 4.75 |
| Helical twist (°) | -0.565 |
| Map sharpening $B$ factor (Å$^2$) | -34 |
| **Model Refinement** | |
| R.M.S deviations | |
| Bond lengths (Å) | 0.002 |
| Bond angles (°) | 0.513 |
| MolProbity score | 1.82 |
| Clash score | 5.83 |
| Rotamer outliers (%) | 0.0 |
| Ramachandran plot | |
| Favored (%) | 92.37 |
| Allowed (%) | 7.63 |
| Outliers (%) | 0.0 |
| EM Ringer score | 3.1 |
| Model vs. Data (CC) | 0.77 |

reconstruction below) as was observed with 263K and aRML prions [5,6]. 2D class averages were obtained from a total of 4449 images (S1 Fig). Cross-over points and multiple axial bands of density in the 2D class averages again showed twisting along the fibril axis, which from the tomography noted above, was left-handed. Fine ribbing perpendicular to the fibril axis was visible in images of 2D class averages with axial spacing consistent with the ~5.0 Å separation between rungs of β-sheets. Finally, 3D classification converged on a single core morphology. In post processing, the pixel size was corrected to reflect a calibrated pixel size of 1.045 Å/pix. With helical reconstruction techniques we obtained ~3.2 Å resolution for much of the a22L fibril core with stacked rungs occurring perpendicular to the fibril axis at a spacing of 4.75 Å (Figs S1B and 2D).

## Atomic model of the a22L prion

We used the final reconstructed 3D density map to build an atomic model of the a22L fibril using the PrP polypeptide sequence known to comprise its protease-resistant core (~81–231 [15]). Parameters of our iterative real and Fourier space refinements and validation were as indicated in Table 1. Residues 94–226 comprised the a22L fibril core, forming major N- and C-terminal lobes (Fig 3B). Similar to the aRML strain [5], and consistent with a22L's deficiency in N-linked glycans and lack of GPI anchors, we failed to see the peripheral densities adjacent to the attachment sites of these post-translational modifications that had been seen previously with wildtype 263K [6] and RML [7] fibrils (compare Figs 2D and 3C: open and blue

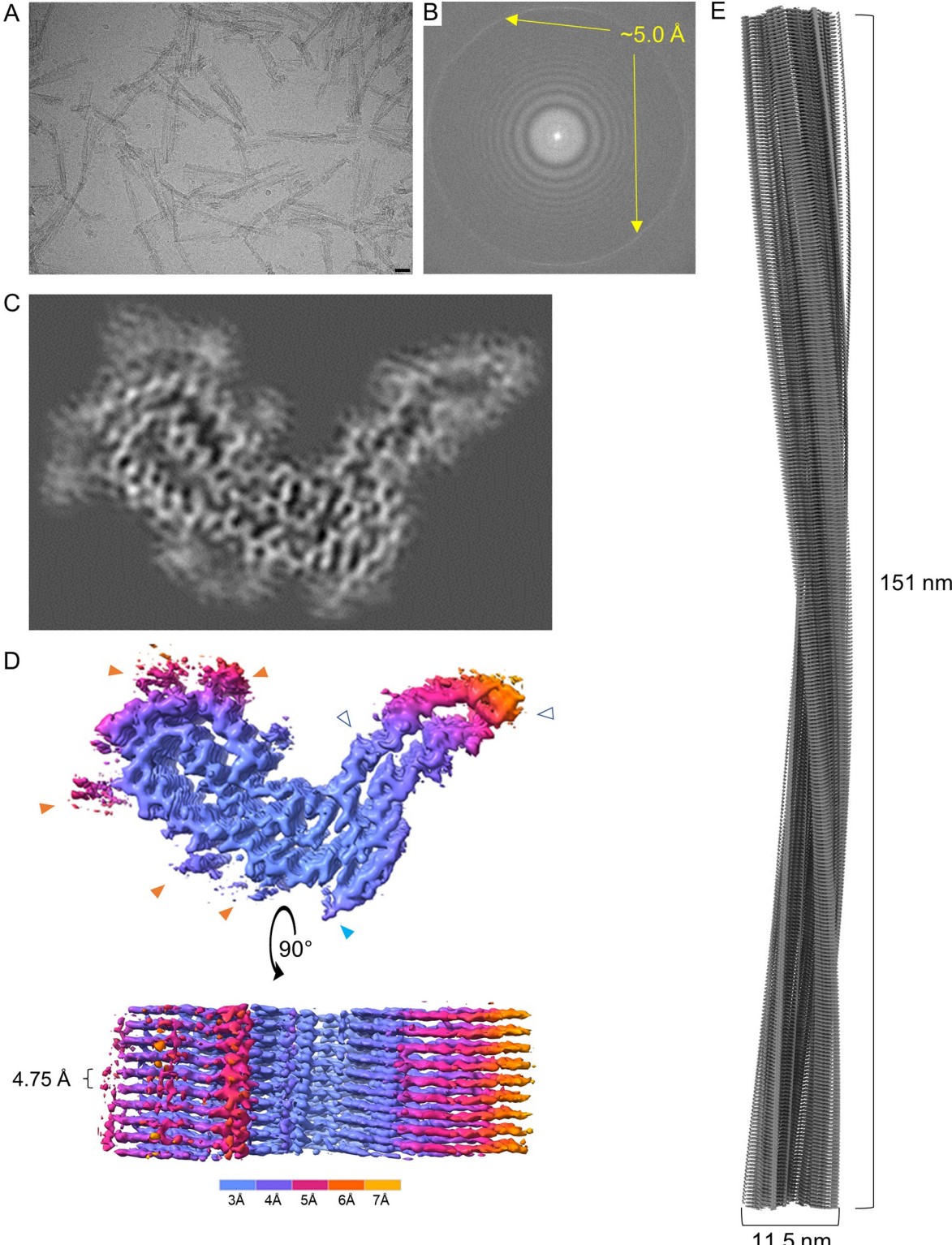

**Fig 2. Cryo-EM images and density maps of a22L prion fibrils.** (A) Representative 2D cryo-EM image of a22L fibrils. Bar = 25 nm. (B) Fast Fourier transform of 2D image in (A) showing signals from regular 5.0 Å spacings (yellow arrows). (C) Projection of density map of fibril cross-section derived from single-particle cryo-EM analysis. (D) Surface depictions of density map with colors showing local resolutions according to color bar. Orange arrowheads: peripheral densities not attributed to polypeptide in subsequent modeling; open arrowheads: the sites of potential N-linked glycans; blue arrowhead: C-terminal site near where GPI anchor would be attached in wild-type prions. (E) Elongated projection of the fibril density map representing a 180˚ twist along the axis (i.e. the cross-over distance).

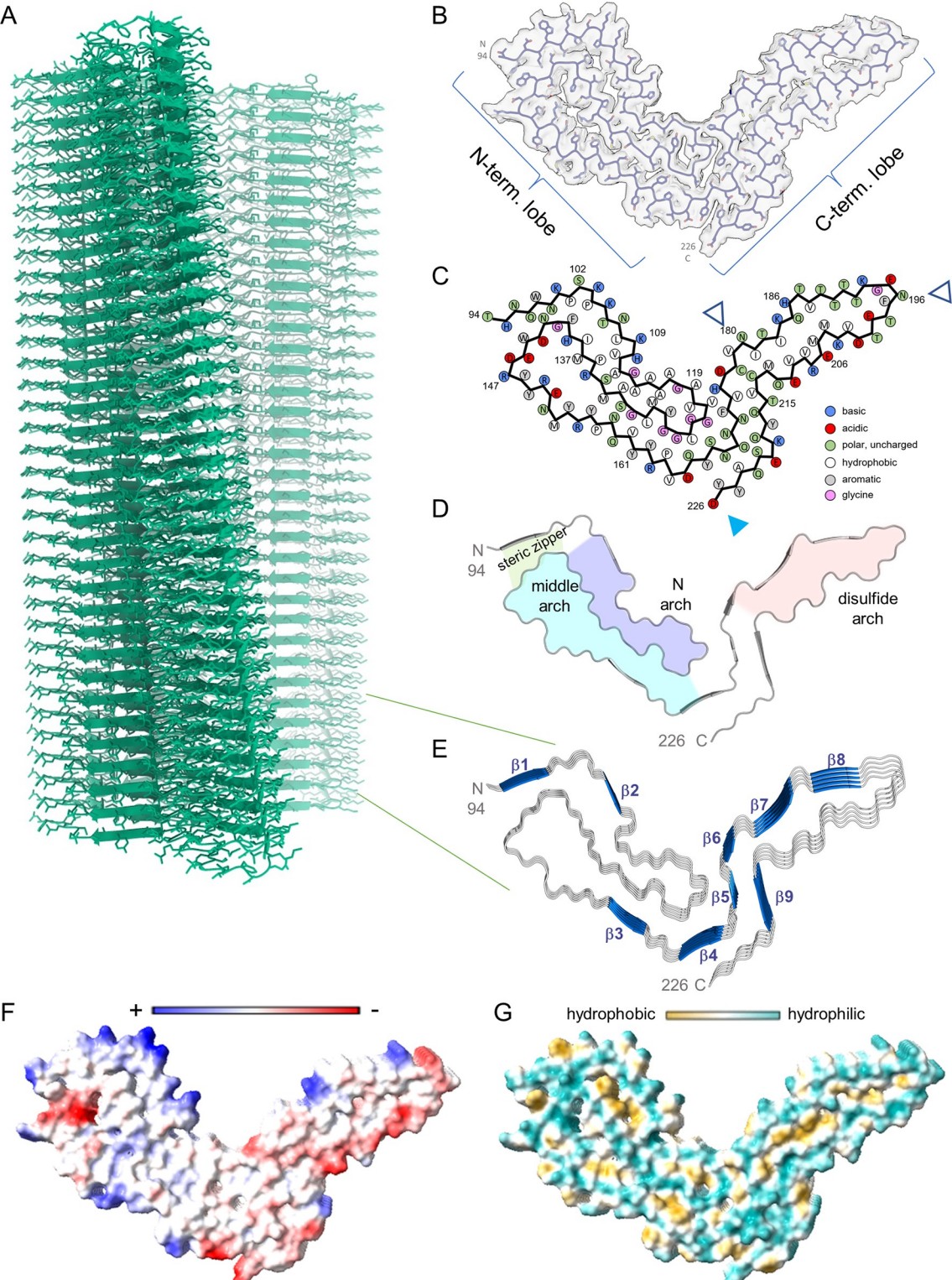

**Fig 3. a22L prion model based on cryo-EM density map.** (A) Extended fibril model as a ribbon diagram. (B) PrP residues 94–226 threaded through a cross-sectional density map. (C) Schematic depiction of fibril core showing side-chain orientations relative to the polypeptide backbone. Open arrowheads mark sites of possible N-linked glycosylation at N180 and N196. The blue arrowhead marks the C-terminus of the ordered core structure (D226), which is near where the GPI anchor would be attached at residue 230 in wild-type, but not anchorless, murine prions. (D) Ribbon cartoon of the fibril cross-section. Structural elements are as labeled. (E) Stacked

pentameric segment of the fibril with β sheets. (F) Coulombic charge representation. (G) Surface hydrophobicity. The analyses in (E-G) were performed using ChimeraX 1.4.

arrowheads). However, consistent with the latter structures, we observed densities adjacent to stacks of cationic side chains on the surface of the N-terminal half of the core (Fig 2D, orange arrowheads, and Fig 3F) that may reflect either remnants of more extreme N-terminal PrP sequence after partial proteolysis or non-PrP ligands.

Several general features of the a22L core were similar to those of the 263K [6], aRML [5], and wtRML [7] structures: a PIRIBS-based architecture; an asymmetric fibril cross-section spanned by a single monomer; N-proximal (N), middle, and disulfide arches; a steric zipper between the N-terminal residues and the head of the middle arch; and preferential exposure of cationic sidechains in the amino-terminal half and anionic sidechains in the C-terminal half of the cross-section (Figs 3 and 4). However, the details of many of these structural features of a22L differ markedly from those of the previously described strains. Some of the divergence between the mouse and hamster prions might be influenced by differences in their respective amino acid sequences at 8 positions within their ordered core regions. Thus, to identify purely conformational determinants of prion strains, we will now focus on how a22L compares to other strains, with emphasis on the aRML strain that was propagated in the same genotype of host.

## Conformational similarities and contrasts between strains

In the N-terminal lobe, the backbones of the a22L and aRML structures can be overlayed closely from residues 94–110 and 133–152 that form the steric zipper between the N-terminal residues and the head of the middle arch as well as the base of the N arch (Fig 5A and 5B). Indeed, this region is also structurally conserved with that of the hamster 263K prion (Fig 5B). However, the respective backbones begin to diverge outside of these spans of residues and into the C-terminal lobe (Fig 5A, 5B and 5D). Among the more notable differences are in the respective heads of the N arch, in which the trans side chain orientations of two pairs of adjacent residues (Val120 and Val121; M128 and L129) are reversed with respect to the polypeptide backbone between the two mouse strains (Fig 5E). These differences are accompanied by major changes in the shape of the N arch heads and their interactions with residues 158–176. The interface between the N- and C-terminal lobes of a22L is less staggered than those of aRML [5], wt RML [7], and 263K [6] prions, allowing a flatter overall shape for the monomers within the PIRIBS stack with the comparison to aRML shown in Fig 5C. The distinctive conformation of a22L residues 168–175 also dictates that their interactions with the C-terminal residues 215–225 also differ from those of aRML with respect to the relative positions of the sidechains of Y168 and Y225 (Fig 5E). The overall shape of the intervening disulfide arch of a22L is also substantially different from aRML [5], wt RML [7], and 263K [6] (Fig 5D). Notably, the a22L and aRML disulfide arches bend in opposite directions, yet both form a more acute angle with the N-terminal lobe than was seen for hamster 263K [6]. According to secondary structure analysis by ChimeraX 1.4 (DSSP) and VMD (STRIDE), both a22L and aRML fibrils have 9 PIRIBS segments. Some are highly conserved between these two mouse prion strains, and even the hamster 263K strain, notably in the N-proximal regions highlighted in Fig 5B. However, they differ markedly elsewhere. Altogether, these results demonstrate both conformationally conserved and divergent features of these rodent prion strains, with the contrasts between a22L and aRML indicating differences that are not attributable to host genotype.

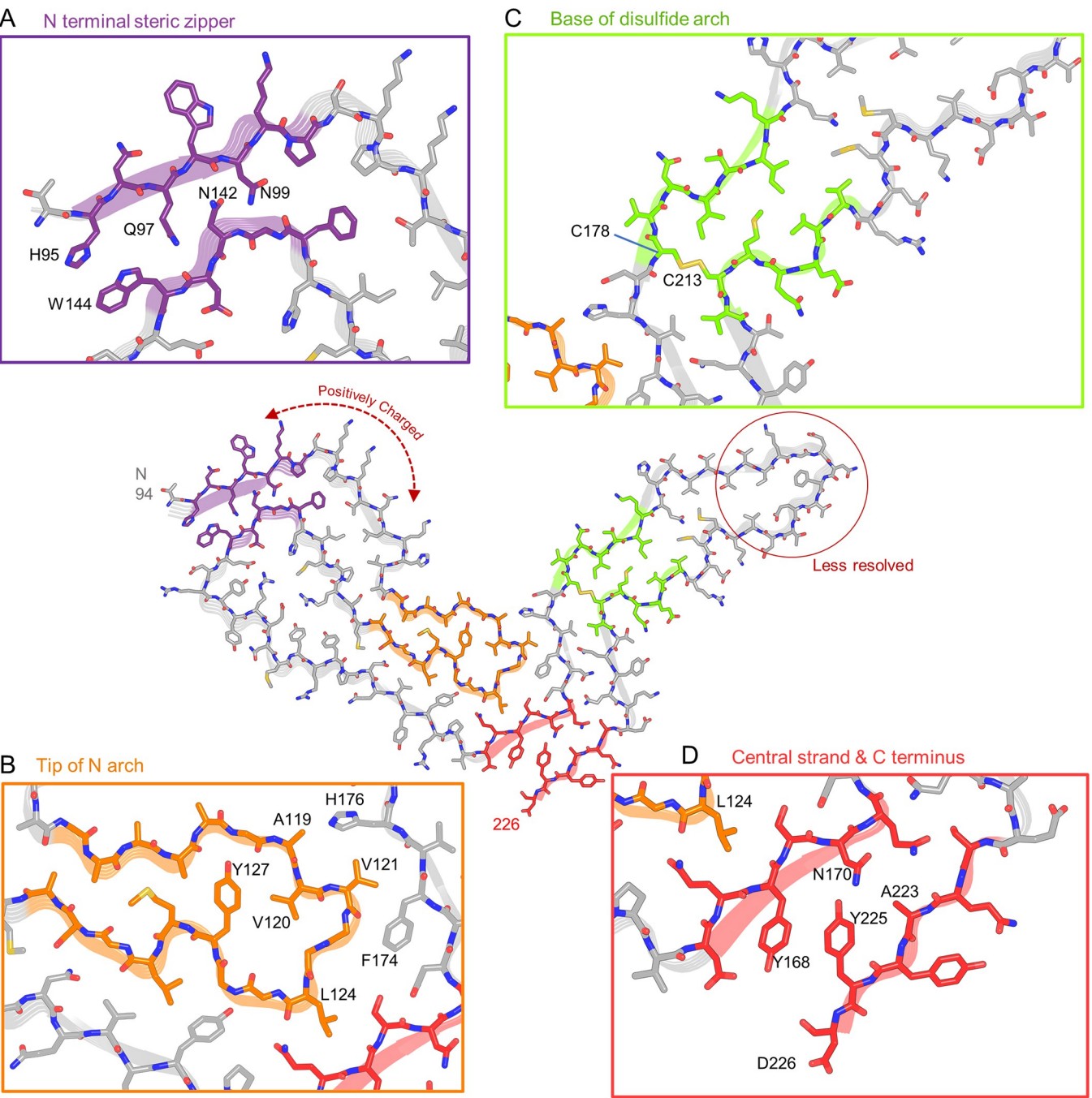

**Fig 4. Features of the a22LL prion core.** (A) Steric zipper (purple) is formed by the N-terminus (H95, Q97, N99) and tip of the middle arch (N142, W144). (B) Tip of the N arch (orange), as well as residues with which they interact across the central interface between the N- and C-terminal lobes. (C) Disulfide bond (C178 –C213) forms the base of disulfide arch (green) that is stabilized by a tight interface adjacent to a widening of the arch toward the tip (central model). The precise side-chain positions in the tip are less certain due to poorer resolution in that portion of the cryo-EM density map (see Fig 2D). (D) Steric zipper (red) between a central β-strand and C-terminus. "Positively charged" indicates a span of mostly basic residues that includes four lysines and a histidine.

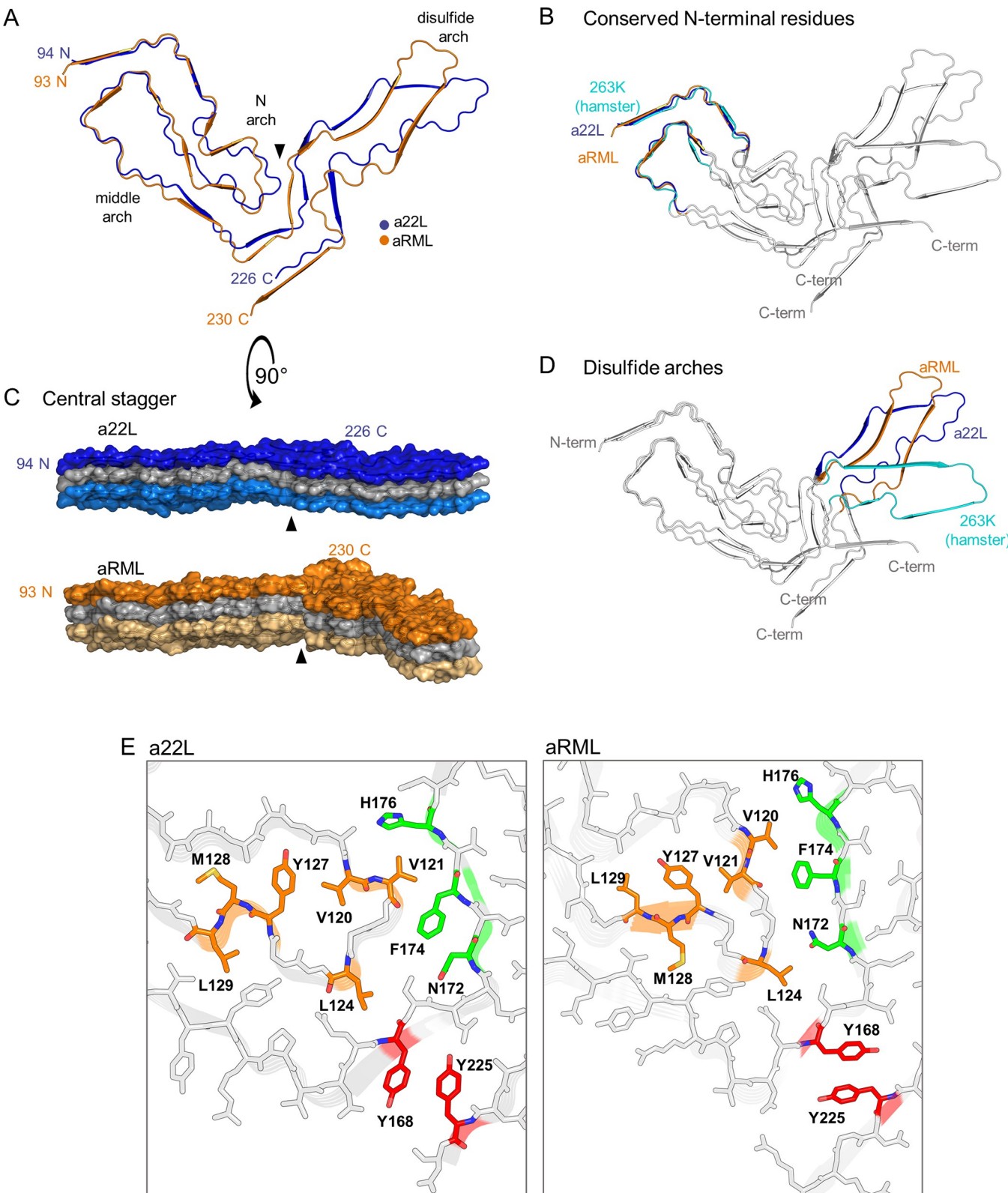

**Fig 5. Comparison of a22L and aRML prions.** (A) Overlay of cross-sections of a22L and aRML cores. (B) Overlay of a22L, aRML and 263K cross-sections, highlighting conformationally conserved N- terminal region. (C) Lateral views of a22L and aRML EM maps (stacks of 3). Arrowheads indicate interface between the head of the N arch and the central strand between the N- and C-terminal lobes of a given monomer, where less axial stagger is seen in a22L than

aRML. (D) Overlay of a22L, aRML and 263K cross-sections highlighting differences in their disulfide arches. (E) Comparison of relative positions of corresponding residues of a22L and aRML within the heads of the N arch (orange) and their interactions with residues (green) across the central (and, in the case of aRML, more staggered) interface between the N- and C-terminal lobes.

## Discussion

### Variations within PIRIBS architectures differentiate prion strains

Although non-fibrillar or sub-fibrillar ultrastructures can accompany amyloid fibrils in infectious brain-derived preparations of prions, all of the high-resolution structures reported to date (n = 5, including a22L described herein) are fibrils with PIRIBS architectures [5–7,25–27]. A much less, if at all, infectious human GSS F198S-associated brain-derived PrP amyloid structure also has a PIRIBS architecture, but with a much smaller order core spanning residues 80–141 (62 residues) [28]. As noted above, the much larger cores (132–138 residues) of the highly infectious prion structures extend nearly to the C-terminus and share several structural motifs; these include an N-proximal steric zipper, N, middle, and disulfide arches, and a central, and often staggered interface between N- and C-terminal lobes [5–7,25–27]. However, variations in these motifs and other features distinguish these rodent prion strains, thereby providing a conformational basis for their consistent propagation as distinct entities through indefinite passages *in vivo* and, ultimately, their characteristic neuropathological and clinical phenotypes.

Our comparison of the a22L structure with our previous structure of the aRML prions propagated in the same genotype of mouse allows us to define conformational strain determinants without the complication of differing PrP amino acid sequences. As detailed above, the N-proximal steric zipper and head of the middle arch are closely similar in the a22L and aRML, as well as the hamster 263K [6], strains (Fig 5B). However, outside of those particular features, the conformations differ markedly in their details (Fig 5A, 5D and 5E). This theme is also reflected in the structure of the mouse wildtype ME7 strain that was recently posted on a preprint server [27], although detailed comparisons await publication of the atomic coordinates of the ME7 model. Perhaps the most obvious of differences between the a22L, aRML and ME7 mouse strains is the overall shapes of their respective disulfide arches, which bend in different directions relative to one another. These murine disulfide arches each differ as well from the hamster 263K disulfide arch [6]. Collectively, these results provide clear near-atomic determination of conformational determinants of prion strains. The comparison of prion strains, e.g. a22L and aRML, produced in a single genotype of mice is most apt in this regard because such prions are derived from mice that produce identical pools of PrP$^C$ substrate, cofactors, and other potential modulators of PrP$^{Sc}$ accumulation.

### Variability in the tips of disulfide arches of prion fibrils

The reduction in resolution toward the tip of the disulfide arch of a22L cryo-EM density map can also be seen in each of the other high-resolution *ex vivo* prion fibril maps (S2 Fig). Although the available resolution for each of these strains is clearly sufficient to establish that their disulfide arches have distinct overall shapes and relative polypeptide backbone maps, the lower resolution at the tips indicates some localized variability in conformation or structure. All of these fibrils have N-linked glycans within the disulfide arches on at least some of the monomers at N180 or N196, but much more so on the wild-type RML and 263K fibrils. Judging from the relative intensities of the glycosylated versus unglycosylated bands on silver-stained gels, our a22L appears to have a glycan on roughly 1/3 of the monomers. Perhaps such structurally and conformationally diverse glycans [29] impose more flexibility in the disulfide arch even if we and others have shown with molecular dynamics simulations that certain

uniform glycans can in theory be accommodated on each rung within a PIRIBS architecture and appear to even stabilize the fibril core [30,31]. Alternatively, there might be more inherent instability in the polypeptide cores in this region of these prion strains that accounts for conformational plasticity. Further studies will be required to determine which of these factors may be most influential. Clearly, among the shared motifs of several *ex vivo* prion structures solved to date, the disulfide arch is one of the most divergent between strains.

## How structural differences might modulate prion strain pathophysiologies

The distinct conformations that we have now seen between a22L (present study) and RML anchorless [5] and wildtype [7] strains, as well as the recently posted wildtype mouse ME7 prion structure [27], indicate that these strains present different surfaces to their tissue environments. Importantly, from the perspective of consistent strain propagation, each infecting strain provides a unique template at the fibril ends that dictates the conformation of newly recruited PrP molecules as they refold and build onto the growing prion fibril [6]. Because the end products have PIRIBS architectures, we expect that a key driving force is hydrogen bonding between the polypeptide backbones of the adjacent monomers; however, further studies will be required to ascertain the conversion mechanism(s) and the factors controlling the template-driven consistency of prion propagation.

In addition to the specific templates on their fibril tips, prion strains display distinctive lateral surfaces in their *in vivo* environments (Fig 6). The parallel in-register alignment of the residues of adjacent monomers in the PIRIBS stack gives linear arrays of cationic, anionic, polar, aromatic, and hydrophobic sidechains. The multivalency of these arrays should enhance the avidity of interactions with complementary polyvalent ligands or molecular assemblies (such as membranes) in their vicinity. Indeed, abundant evidence indicates key roles for various cofactors such as sulfated glycans, nucleic acids, and phospholipids in PrP conversion and infectious prion propagation [32–36]. Among the apparent roles of such cofactors is to

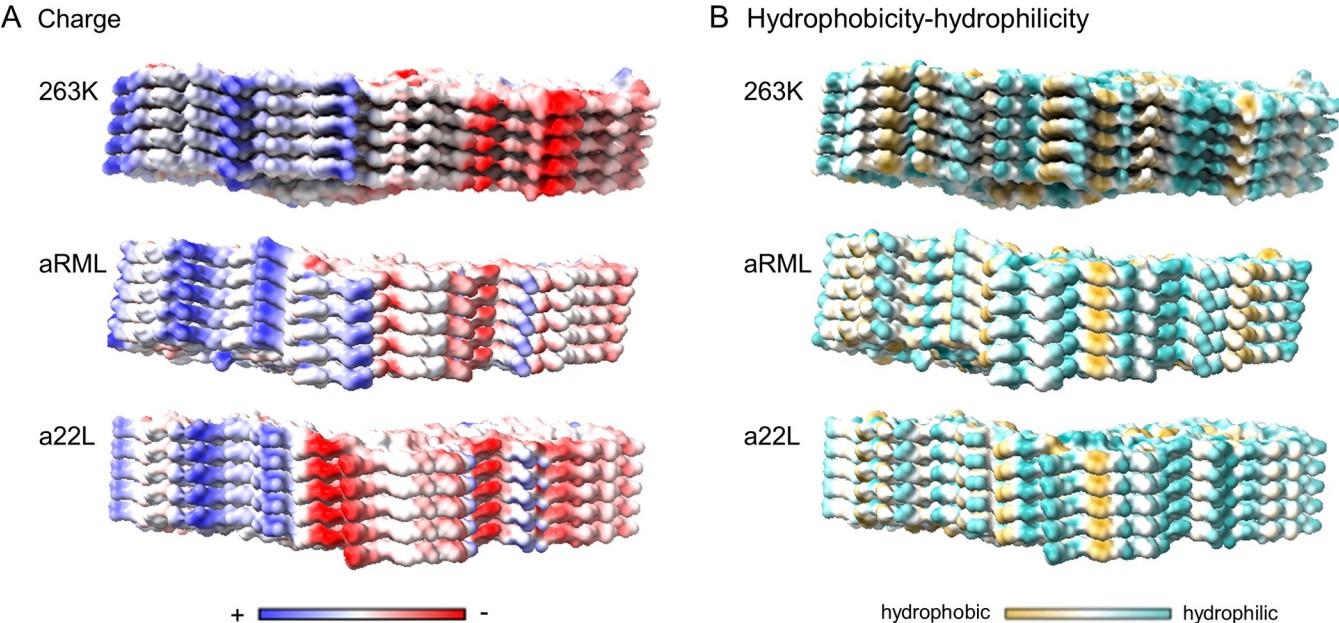

**Fig 6. Strain-dependent distributions of coulombic charges and surface hydrophobicity.** (A) Coulombic charge distributions in 263 K (PDB: 7LNA) [6], aRML (PDB: 7TD6) [5] and a22L (PDB: 8EFU) prion strains. (B) Hydrophobicity distributions. These analyses were performed in ChimeraX 1.4.

mitigate the electrostatic repulsion between stacked in-register cationic or anionic residues [37–41]. The efficacy of such cofactors can be prion strain-dependent [42], and likely depends on their relative abilities to interact optimally with prion surfaces and/or conversion intermediates to facilitate fibril growth. At least some such potential cofactors include sulfated glycosaminoglycans, membranes, extracellular matrix components, and chaperones. These types of molecules can differ between cell types or brain regions. Thus, they might interact preferentially with certain prion strains over others due to their respective chemical surfaces and control where within tissues a given strain optimally propagates and accumulates. This, in turn, likely contributes to the well-known strain-dependent patterns of PrP$^{Sc}$ accumulation and pathological lesions in the brain.

Strain-dependent conformational differences might also serve as a basis for differences in binding by complement factors [43,44] and other elements of innate immune and protein quality control mechanisms that might be involved in PrP$^{Sc}$ degradation or neuroinflammation. The neurotoxic mechanisms of prion disease are not well understood, but it is likely that conformational differences between strains modulate how prions interact with other factors, including PrP$^{C}$, that are involved either directly, or indirectly, with neuronal and glial dysfunction.

## Conclusions

Our a22L structure allows us to provide high-resolution structural evidence for the long-proposed hypothesis that prion strains from a given type of host differ in conformation [9–13]. This structure also adds to the short list of brain-derived pathological prions [5–7,25–27] and PrP amyloids [28] that have PIRIBS-based fibril architectures, as we had predicted some years ago based on our observations of PIRIBS backbones in PrP$^{Sc}$-seeded synthetic PrP fibrils [38]. The highly infectious prion structures also share several key structural motifs, but the conformational details of these motifs vary between strains, even when the fibrils are assembled from monomers of the same PrP amino acid sequence. The structures of many mammalian prion strains remain to be determined, so it is premature to assume that all will share PIRIBS architectures or structural motifs, especially considering the fundamentally non-fibrillar 2D crystalline arrays that have been observed in *ex vivo* preparations of infectious prions [45,46]. Moreover, particles that are too short to be described as fibrillar have been found in PrP$^{Sc}$ or other prion preparations [14,17,47,48] but further studies are required to determine whether such particles share the same core structure as more elongated fibrils, or are fundamentally distinct oligomeric structures. Nonetheless, it is clear that amyloid fibrils of PrP$^{Sc}$ accumulate *in vivo* and are the main component of many highly purified and infectious *ex vivo* prion preparations. The evidence so far shows that such fibrillar forms of PrP$^{Sc}$ have strain-specific PIRIBS-based cores. This conformational diversity provides a molecular basis for the existence of prion strains. When considered in the larger context of the many diseases caused by the accumulation of pathological protein amyloids, these data reinforce the concept that variations in the core conformations of self-propagating amyloids can underpin marked differences in disease phenotype.

## Materials and methods

### a22L fibril purification

a22L fibrils were purified from brains of transgenic expressing only GPI anchorless PrP obtained under Rocky Mountain Laboratories Animal Care and Use Committee Protocol #03–34.1, and the preparation was characterized as described in previous reports [15,16]. Just prior to cryo-EM grid preparation, the fibril preparations were vortexed and allowed to sit for

several minutes to pellet highly bundled fibrils. Aliquots from the supernatant fraction were diluted in 20 mM Tris pH 7.4, 100 mM NaCl, vortexed, and then supplemented with 0.02% amphipol 8–35 and sonicated immediately prior to grid preparation.

## Bioassay of purified a22L prion strains in Tg mice

All mice were housed at the Rocky Mountain Laboratory in an AAALAC accredited facility in compliance with guidelines provided by the Guide for the Care and Use of Laboratory Animals (Institute for Laboratory Animal Research Council). Experimentation followed Rocky Mountain Laboratory Animal Care and Use Committee approved protocol 2021-011-E. To estimate the infectivity of the purified a22L preparation, groups of 4–6 tga20 homozygous mice [22] were anesthetized with isoflurane and injected in the left-brain hemisphere with 0.1 μg of purified a22L prions diluted in 30 μl phosphate buffered balanced saline solution + 2% fetal bovine serum. Following inoculation, mice were monitored for onset of prion disease signs and euthanized when they displayed signs of prion disease including ataxia, flattened posture, delayed response to stimuli, and somnolence.

## Negative stain electron microscopy

Three μl of sample was added to glow-discharged ultrathin carbon on lacey carbon support film grids (400 mesh, Ted Pella, Redding, CA) for one min, briefly washed with $dH_20$, then negatively stained with Nano-W (2% methylamine tungstate) stain (Nanoprobes, Yaphank, NY). Grids were imaged at 80 kV with a Hitachi HT-7800 transmission electron microscope and an XR-81 camera (Advanced Microscopy Techniques, Woburn, MA).

## Cryo- EM Grid preparation

Grids (C-Flat 1.2/1.3 300 mesh copper grids, Protochips, Morrisville, NC) were glow-discharged with a 50:50 oxygen/hydrogen mixture in a Solarus 950 (Gatan, Pleasanton, CA) for 10 s. Grids were mounted in an EM GP2 plunge freezer (Leica, Buffalo Grove, IL) and a 3 μl droplet of 0.02% amphipol A8-35 in phosphate buffered saline was added to the carbon surface and hand blotted to leave a very thin film. The tweezers were then raised into the chamber of the plunge freezer, which was set to 22˚C and 90% humidity. A recently cuphorn-sonicated 3 μl of sample was added to the carbon side of the grid and allowed to sit for 60 s. The sample was subsequently blotted for ∼4 s followed by a 3 s drain time before plunge freezing in liquid ethane kept at −180˚C. Grids were mounted in AutoGrid assemblies.

## Cryo-electron tomography

Grids were loaded into a Krios G1 (Thermo Fisher Scientific, Waltham, MA) transmission electron microscope operating at 300 kV with a K3 (Gatan, Pleasanton, CA) and a Bioconti-nuum GIF (Gatan, Pleasanton, CA) with a slit width of 20 eV. Tilt series were acquired using SerialEM [49] at a 2.178 Å pixel size at ±60˚, 2˚ increment in a dose symmetric manner around 0˚ [50] with defocus values ranging from −3 to −6 μm and a total dose of ~170 e⁻/Å². Tomograms were reconstructed and 16 were analyzed using IMOD [51]. To verify that our imaging system preserved handedness, we negatively stained bacteria onto a finder grid and acquired tilt-series of an asymmetric letter to confirm orientation did not change during imaging nor through tomographic reconstruction. We then used the bacteria as fiducials to confirm that there were only rotation changes during magnification increases from the tilt-series magnification of the finder grid letter to the tilt-series magnification of a22L.

### Single particle acquisition

Grids were loaded into a Krios G4 (Thermo Fisher Scientific, Waltham, MA) transmission electron microscope operating at 300 kV equipped with a K3 (Gatan, Pleasanton, CA) and a Biocontinuum GIF (Gatan, Pleasanton, CA) operating at a slit width of 20 eV. 6159 gain normalized movies were acquired at a pixel size of 1.1 Å/pix using EPU (Thermo Fisher Scientific, Waltham, MA). Total dose per movie was 57 e⁻/Å² in CDS mode with a dose rate of 8 e⁻/pixel/second. Defocus values were set to cycle multiple steps between -0.5 and 2 μm.

### Image processing

Motion correction of raw movie frames was performed with RELION 4.0 beta [24]. CTF estimation was performed using CTFIND4.1 [52]. All subsequent processing was performed in RELION. Fibrils were picked manually, and segments were extracted with an inter-box distance of 14.7 Å using box sizes of 768 and 384 pixels. Segments from the large box size were used to estimate the fibril cross-over distance of the fibril for estimating initial twist parameters. 2D classes from the short segments were used to generate an initial 3D model. Reference-free 2D class averaging was performed, using a regularization parameter of $T = 2$, a tube diameter of 160 Å, and the translational offset limited to 4.9 Å. The initial model was used for 3D auto refinement with C1 symmetry, initial resolution limit of 40 Å, initial angular sampling of 7.6°, offset search range of 5 pixels, initial helical twist of −0.65°, initial helical rise of 4.9 Å, and using 50% of the segment [53] central Z length. 3D classification was performed without allowing for image alignment. Classes displaying well aligned segments, from auto refinement, were selected for further refinement. Refinement of the helical twist and rise resulted in a twist of −0.565° and rise of 4.997 Å. Iterative cycles of CTF refinement and Bayesian polishing were used until resolution estimates stabilized. Error in the detector pixel size was corrected for in post processing. The pixel size was adjusted to 1.045 Å/pix to yield a subunit rise of 4.75 Å [53]. Post processing in RELION was performed with a soft-edged mask representing 10% of the central Z length of the fibril. Resolution estimates were obtained between independent refined half-maps at 0.143 FSC.

### Model refinement

The a22L atomic model was built starting from the aRML model with real-space refinement and manual editing being performed, using Coot [54], to build into the EM density. Residues comprising the protease-resistant core (i.e. ~94–226) were included in the model. Individual subunits were translated to generate a stack of five consecutive subunits. The subunits were rigid-body fit in ChimeraX [55] for initial placement. Real-space refinement using Phenix [56,57] and Fourier space refinement using RefMac5 were performed iteratively with subsequent validation. Model validation was performed with CaBLAM [58], MolProbity [59], and EMringer [60], and any identified outliers/clashes were corrected for subsequent iterative refinements/validation.

### Supporting information

**S1 Fig.** (A) Representative 2D class averages showing lateral views of a22L fibril segments. The enlargement of the class average boxed in green allows easier visualization of the fine horizontal bands running perpendicular to the fibril axis. (B) Fast Fourier transform of 2D class average boxed in green indicating signals at ~5.0 Å. (C) Fourier shell correlation plots of masked and unmasked models.
(PDF)

**S2 Fig. Surface depictions of density map cross-sections of 263K [6] and aRML [5] prion fibrils with colors showing local resolutions according to the color bar.** Compare to a22L in Fig 2D.
(PDF)

## Acknowledgments

We thank Ms. Elizabeth Fisher for helpful suggestions and oversight of the NIH EM facility; Jeff Severson for assistance with animal work. We thank Drs. Sonja Best, Suzette Priola, and Cathryn Haigh for critical internal review of this manuscript. This work utilized the computational resources of the NIH HPC Biowulf cluster (http://hpc.nih.gov).

## Author Contributions

**Conceptualization:** Gerald S. Baron, Allison Kraus, Byron Caughey.

**Data curation:** Forrest Hoyt, Cindi L. Schwartz.

**Formal analysis:** Forrest Hoyt, Efrosini Artikis, Cindi L. Schwartz, Allison Kraus.

**Funding acquisition:** Byron Caughey.

**Investigation:** Forrest Hoyt, Parvez Alam, Efrosini Artikis, Cindi L. Schwartz, Andrew G. Hughson, Brent Race, Chase Baune, Gregory J. Raymond, Gerald S. Baron.

**Methodology:** Forrest Hoyt, Parvez Alam, Efrosini Artikis, Cindi L. Schwartz, Andrew G. Hughson, Gerald S. Baron.

**Project administration:** Byron Caughey.

**Resources:** Parvez Alam, Andrew G. Hughson, Gregory J. Raymond, Gerald S. Baron.

**Software:** Forrest Hoyt, Efrosini Artikis, Cindi L. Schwartz.

**Supervision:** Brent Race, Byron Caughey.

**Validation:** Forrest Hoyt, Efrosini Artikis, Cindi L. Schwartz, Allison Kraus.

**Visualization:** Forrest Hoyt, Efrosini Artikis, Cindi L. Schwartz, Byron Caughey.

**Writing – original draft:** Forrest Hoyt, Parvez Alam, Efrosini Artikis, Cindi L. Schwartz, Andrew G. Hughson, Byron Caughey.

**Writing – review & editing:** Forrest Hoyt, Parvez Alam, Efrosini Artikis, Cindi L. Schwartz, Andrew G. Hughson, Brent Race, Gerald S. Baron, Allison Kraus, Byron Caughey.

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
