## [Decision Letter · Decision Letter 0]

14 Oct 2022

Dear Dr. Caughey,

Thank you very much for submitting your manuscript "Cryo-EM of prion strains from the same genotype of host identifies conformational determinants" for consideration at PLOS Pathogens. As with all papers reviewed by the journal, your manuscript was reviewed by members of the editorial board and by several independent reviewers. The reviewers appreciated the attention to an important topic. Based on the reviews, we are likely to accept this manuscript for publication, providing that you modify the manuscript according to the minor recommendations outlined in the Reviewers' comments below.

Sincerely,

Amanda L Woerman

Associate Editor

PLOS Pathogens

Neil Mabbott

Section Editor

PLOS Pathogens

Kasturi Haldar

Editor-in-Chief

PLOS Pathogens

orcid.org/0000-0001-5065-158X

Michael Malim

Editor-in-Chief

PLOS Pathogens

orcid.org/0000-0002-7699-2064

Reviewer Comments (if any, and for reference):

Reviewer's Responses to Questions

**Part I - Summary**

Reviewer #1: This manuscript describes the structures of two PrPSc prion strains with the same PrP sequence at a near atomic resolution. Such structures have been resolved using cryo-electron microscopy (cryo-EM). This is of great importance to the prion field as it definitively solves the issue of the molecular underpinnings of prion strains. From this study, it is clear that different PrPSc prions consist of slightly different PIRIBS architectures of the PrP sequence -variations on the general common theme. Such conclusion had already been hinted at from previous resolved PrPSc structures: mouse RML PrPSc, and hamster 263K PrPSc. However, the fact that these strains feature different PrP sequences left, theoretically at least, some possible uncertainty, which is definitively set to rest with the current study.

The Authors have a solid track record in the field, having solved the structures of 263K, and anchorless RML PrPSc. The current study is technically straightforward and presentation and interpretation of the data, impeccable. I might point out to a number of minutiae that in my opinion might be improved, but I believe that in the interest of a quick decision can be perfectly avoided.

Reviewer #2: A strong and well-written manuscript describing the conformational differences of two mouse prion strains I think it's vitally important that this research be published, even though the results were consistent with what was expected based on recent literature in the field and first principles.

This field is moving very quickly, thus it's vital that research like this is published quickly to help lay a solid foundation for this field.

Reviewer #3: This study shows the structural analysis of a murine GPI-anchorless prion strain, anchorless 22L, at 3.2 Å resolution and compares the structure with the previously resolved anchorless RML. The fibril analysis is robust and the fibrils rigorously characterized. This work is a significant advance for the field and answers a long standing question on strains, as it illustrates key differences between the structures using two experimental prion strains that have been studied in vitro and in vivo. Notably, a major structural difference between two strains is revealed in the disulfide β-arch.

**Part II – Major Issues: Key Experiments Required for Acceptance**

Reviewer #1: None

Reviewer #2: none noted

Reviewer #3: (No Response)

**Part III – Minor Issues: Editorial and Data Presentation Modifications**

Reviewer #1: None (vide supra).

Reviewer #2: For Figure 2, the arrowheads have a fairly thick shape outline and the color differences weren't quickly obvious until I moved the image to a big screen. In addition, they may be difficult for someone who is RG colorblind.

line 324-328: it's hypothesized that the glycans can impose flexibility in the disulfide arch. it's also stated that the top third of the arch has locally low/moderate resolution of 5-7 Angstroms. This makes sense, but the previous paragraph (308-310) that the different disulfide arch conformations are the "most obvious differences between the ... strains." I don't doubt for a second that these are different strains, but it would be extremely helpful if the authors would help with the interpretation of the differences for those who are not structural biologists. if the conformational of the arches are flexible and not well resolved, how much confidence should the reader have that the three strains are dramatically different in this particular area? what is the most scientifically valid manner in which to interpret the poor resolution and conformational flexibility when assessing the conformational differences?

A second comment on lines 324-328: it's stated that the glycans may impose flexibility to the disulfide arch. but then in the same sentence it's stated that uniform glycans can stabilize the fibril core. This argument requires some additional explanation, as it seems anti-thermodynamic. if the glycans impose additional entropy to the system in the form of conformational flexibility, how can they make it more stable? Then in the next sentence it discusses the inherent instability of the polypeptide core in the arch. please clarify.

Very minor points that don't require modification;

Line 365: "This, in turn, might contribute to the well-known strain PrPSc dependent patterns of accumulation and pathological lesions in the brain." and the similar, line 379: "it seems plausible that conformational differences between the strains could modulate how prions interact with other factors". I think the conclusions of this manuscript are strong enough that it's difficult reach any other conclusion". It seems more than just plausible. This is a statement of the 'strain hypothesis', correct?

Paragraph starting line 339: This paragraph contains generalized speculation. The difference in charge distribution between the strains is mentioned multiple times. Trying to read between the lines, do the authors believe that the location and concentration of surface charged residues could lead to divergent biology for the strains? This seems plausible, but leads to the unanswered question as to how that might actually work.

Reviewer #3: I have minor suggestions.

1) A brief description of fibril variability would be important to include. Was there heterogeneity in fibril structures? Were there any short fibrils examined?

2) Were there differences in the presence, size, and location of non-protein molecules bound to fibrils within a strain, and between strains?

3) Perhaps the authors could speculate on why the differences in structures are concentrated at the disulfide β-arch.

4) The authors may consider further emphasizing the importance of the work to the amyloid field in the Discussion.

PLOS authors have the option to publish the peer review history of their article (what does this mean?). If published, this will include your full peer review and any attached files.

Reviewer #1: **Yes: **Jesús R. Requena

Reviewer #2: No

Reviewer #3: No

Figure Files:

Data Requirements:

Reproducibility:

References:

---

## [Editor Report · Decision Letter 1]

24 Oct 2022

Dear Dr. Caughey,

We are pleased to inform you that your manuscript 'Cryo-EM of prion strains from the same genotype of host identifies conformational determinants' has been provisionally accepted for publication in PLOS Pathogens.

Best regards,

Amanda L. Woerman

Associate Editor

PLOS Pathogens

Neil A. Mabbott

Section Editor

PLOS Pathogens

Kasturi Haldar

Editor-in-Chief

PLOS Pathogens

orcid.org/0000-0001-5065-158X

Michael Malim

Editor-in-Chief

PLOS Pathogens

orcid.org/0000-0002-7699-2064
---

## [Editor Report · Acceptance letter]

2 Nov 2022

Dear Dr. Caughey,

We are delighted to inform you that your manuscript, "Cryo-EM of prion strains from the same genotype of host identifies conformational determinants," has been formally accepted for publication in PLOS Pathogens.

Best regards,

Kasturi Haldar

Editor-in-Chief

PLOS Pathogens

orcid.org/0000-0001-5065-158X

Michael Malim

Editor-in-Chief

PLOS Pathogens

orcid.org/0000-0002-7699-2064